# Frankincense (*Boswellia serrata*) Extract Effects on Growth and Biofilm Formation of *Porphyromonas gingivalis,* and Its Intracellular Infection in Human Gingival Epithelial Cells

**David Vang** [1]**, Aline Cristina Abreu Moreira-Souza** [1]**, Nicholas Zusman** [2]**, German Moncada** [1]**, Harmony Matshik Dakafay** [1]**, Homer Asadi** [1]**, David M. Ojcius** [1] **and Cassio Luiz Coutinho Almeida-da-Silva** [1,*]

1   Department of Biomedical Sciences, Arthur A. Dugoni School of Dentistry, University of the Pacific, San Francisco, CA 94103, USA; dvang@pacific.edu (D.V.); asouza1@pacific.edu (A.C.A.M.-S.); gmoncada@pacific.edu (G.M.); hmatshikdakafay@pacific.edu (H.M.D.); hasadi@pacific.edu (H.A.); dojcius@pacific.edu (D.M.O.)
2   Dental Surgery Program, Arthur A. Dugoni School of Dentistry, University of the Pacific, San Francisco, CA 94103, USA; n_zusman@u.pacific.edu
*   Correspondence: csilva2@pacific.edu

**Abstract:** Frankincense is produced by *Boswellia* trees, which can be found throughout the Middle East and parts of Africa and Asia. *Boswellia serrata* extract has been shown to have anti-cancer, anti-inflammatory, and antimicrobial effects. Periodontitis is an oral chronic inflammatory disease that affects nearly half of the US population. We investigated the antimicrobial effects of *B. serrata* extract on two oral pathogens associated with periodontitis. Using the minimum inhibitory concentration and crystal violet staining methods, we demonstrated that *Porphyromonas gingivalis* growth and biofilm formation were impaired by treatment with *B. serrata* extracts. However, the effects on *Fusobacterium nucleatum* growth and biofilm formation were not significant. Using quantification of colony-forming units and microscopy techniques, we also showed that concentrations of *B. serrata* that were not toxic for host cells decreased intracellular *P. gingivalis* infection in human gingival epithelial cells. Our results show antimicrobial activity of a natural product extracted from *Boswellia* trees (*B. serrata*) against periodontopathogens. Thus, *B. serrata* has the potential for preventing and/or treating periodontal diseases. Future studies will identify the molecular components of *B. serrata* extracts responsible for the beneficial effects.

**Keywords:** *Porphyromonas gingivalis*; *Fusobacterium nucleatum*; *Boswellia serrata*; biofilm; epithelial cells

## 1. Introduction

There is growing awareness that over-prescription and misuse of antibiotics by health-care providers are major contributors to the antibiotic resistance of human pathogens. This leads to an increasing demand for novel antimicrobials to treat infections caused by different pathogens. The synthesis of new synthetic antibiotics is expensive and complex, and the average time from discovery to market for a general broad-spectrum antibiotic is around 14 years [1]. The need for sustainable, safe, and cost-effective therapeutics is increasing. In this context, natural products can reduce the likelihood of antibiotic resistance and, having been used for centuries by humans, are usually considered to be less toxic. Some natural products are believed to enhance the body's immune response, aiding in the natural defense against infections [2].

Frankincense is a gum resin that is a byproduct of incisions made to the trunks of the *Boswellia* tree [3,4]. The *Boswellia* genus comprises around thirty different species and is found in arid regions of Africa, the Arabian Peninsula, and South Asia [4]. Frankincense is primarily obtained from *B. frereana*, *B. sacra*, *B. papyrifera*, and *B. serrata*. These specific species are indigenous to Somalia, Yemen, Oman, and parts of India and Pakistan [4,5].

For centuries, frankincense has been used as a traditional medicine for constipation and inflammatory diseases as well as incense for religious rituals and funeral ceremonies [6]. Frankincense has been attracting more attention due to its potential effects against inflammation, cancer, diabetes, and microbial infection [4,7,8].

Periodontitis is a chronic inflammatory disease affecting the supporting structures around teeth (gingiva, periodontal ligament, and alveolar bone), and can ultimately lead to uncontrolled bone resorption and irreversible tooth loss [9]. Periodontitis affects approximately 20–50% of the worldwide population [10], and has been linked with several systemic diseases such as cardiovascular disease, diabetes, respiratory tract infections, cancer, and neurodegenerative disorders [11]. Two oral pathogens that have been directly linked to periodontitis include the Gram-negative anaerobic bacteria *Porphyromonas gingivalis* and *Fusobacterium nucleatum* [9,12,13]. Interestingly, *P. gingivalis* and *F. nucleatum* use different mechanisms to modulate the host immune response and contribute to biofilm formation and dissemination [13–16]. *P. gingivalis* and *F. nucleatum* are intracellular pathogens and have been described to evade the host immune response via several virulence factors [9,14,16–22].

Given the rise in the antibiotic resistance of human pathogens, it is important to develop effective antibiotic-independent treatments to eliminate periodontopathogens in the oral cavity. The use of natural products can provide a safe alternative to antibiotics to treat chronic inflammation without compromising the host.

As we previously reviewed, different frankincense extracts and compounds have been reported to have antimicrobial effects against different pathogens [4]. Essential oils from *B. carterii* exhibited antimicrobial activity against several Gram-positive and Gram-negative bacteria, such as *Bacillus subtilis*, *B. circulans*, *Streptococcus faecalis*, and *Escherichia coli*, as well as against two fungal pathogens, *Candida albicans* and *Saccharomyces cerevisiae* [23]. Other studies explored the in vitro [24,25] and in vivo [24] antimicrobial effects of *B. sacra* against multi-resistant strains of *Staphylococcus aureus* and *Pseudomonas aeruginosa*. So far, only two studies investigated the effects of frankincense against oral microbes [26,27]. Raja et al. screened several oral pathogens but focused their in vitro experiments on *S. mutans* [26]. Attalah et al. described the antibacterial effects of *B. sacra* on the growth and survival of planktonic bacteria and on the formation of biofilm using *P. gingivalis* clinical isolates [27]. However, no studies have explored the effects of *B. serrata* as an antimicrobial that can inhibit biofilm formation and intracellular infection of oral cells by *P. gingivalis* and *F. nucleatum*.

In this study, we explored the potential effects of *B. serrata* extracts against *P. gingivalis* and *F. nucleatum* growth in vitro, and infection in human epithelial oral cells. We describe, for the first time, the antimicrobial effects of *B. serrata* on biofilm formation, biofilm reduction and intracellular infection by oral pathogens. We found that frankincense can decrease *P. gingivalis* growth and biofilm formation and inhibit intracellular infection by *P. gingivalis* in human oral cells.

## 2. Materials and Methods

### 2.1. Frankincense (B. Serrata Extract)

*B. serrata* extract USP reference standard (cat# 1076250, Sigma-Aldrich—St. Louis, MO, USA) was isolated from *B. serrata* trees from India. The obtained extract was resuspended in dimethyl sulfoxide (DMSO; Sigma-Aldrich—St. Louis, MO, USA) at 128 mg/mL and kept at 4 °C until use.

### 2.2. Bacterial Strains, Human Oral Cells, and Growth Conditions

*P. gingivalis* (ATCC® 33277) and *F. nucleatum* (ATCC® 25586) were purchased from American Type Culture Collection (ATTC—Manassas, Virginia, Washington, DC, USA) and grown as previously described [28]. Briefly, *P. gingivalis* and *F. nucleatum* were separately grown anaerobically at 37 °C for approximately 7 days in *Brucella* agar plates (Anaerobe systems, cat# AS-141—Morgan Hill, CA, USA). Isolated and pure colonies were collected from

agar plates containing either *P. gingivalis* or *F. nucleatum* to be inoculated in supplemented BHI broth at 37 °C for approximately 48 h under anaerobic conditions. *P. gingivalis* and *F. nucleatum* broth were prepared as we previously described [28]. Freshly grown bacteria were used for experiments after being collected at the log phase and quantified by optical density at 600 nm in the SpectraMax iD3 microplate reader (Molecular Devices—Ramsey, MN, USA).

Immortalized human gingival keratinocytes (HPV-16GM), referred to as gingival epithelial cells (GECs), were obtained from Applied Biological Materials (ABM, cat#T0717—Richmond, CA, USA) and maintained as we previously described [29]. Briefly, GECs were grown and maintained in keratinocyte-serum-free medium supplemented with 30 μg/mL of bovine pituitary extract, 0.2 ng/mL of human recombinant epidermal growth factor, 100 U/mL of penicillin, and 100 μg/mL of streptomycin (Gibco—Gaithersburg, MO, USA). The cells were grown in a humidified incubator, at 37 °C, 5% $CO_2$, and quantified using trypan blue (Sigma-Aldrich—St. Louis, MO, USA) exclusion before each experiment.

### 2.3. Minimum Inhibitory Concentration (MIC)

*P. gingivalis* and *F. nucleatum* were grown in broth at 37 °C for approximately 48 h prior to the experiment. In a 96-well tissue culture plate (Costar, Corning—Glendale, AZ, USA), a serial dilution was performed from the highest *B. serrata* extract concentration of 512 μg/mL to the lowest concentration of 0.25 μg/mL, using bacterial broth in duplicates. Freshly grown *P. gingivalis* or *F. nucleatum* were added to the wells at a final concentration of $5 \times 10^5$ CFU/mL. The plates were incubated under anaerobic conditions at 37 °C. After 48 h, the plates were examined for growth and turbidity under the different concentrations of frankincense extract by measuring the optical density at 600 nm in the SpectraMax iD3 microplate reader.

### 2.4. Biofilm Formation Assay

*P. gingivalis* or *F. nucleatum* biofilm formation was determined using the crystal violet staining assay [30], and adapted to our experiments from previous studies [31]. Biofilm formation assays were performed by adding a serial dilution of *B. serrata* extract at different final concentrations (128, 64, 32, 16, 8, 4, 2, 1, 0.5, 0.25 μg/mL) in duplicates in 24-well plates (Costar, Corning—Glendale, AZ, USA). Freshly grown *P. gingivalis* or *F. nucleatum* were added to all wells at a final concentration of $1 \times 10^7$ CFU/mL. Antibiotics (100 U/mL of penicillin and 100 μg/mL of streptomycin, from Gibco - Gaithersburg, MO, USA) were used as positive control for bacterial growth inhibition. Bacterial broth was used as negative control. After 48 h of incubation at 37 °C and under anaerobic conditions, the supernatants were discarded, and non-adherent bacteria were gently rinsed off with sterile phosphate-buffered saline (PBS, Gibco—Gaithersburg, MO, USA). Adhered biofilms were fixed with cold absolute methanol for 15 min at room temperature. Then, the biofilms were gently washed once with PBS to remove the methanol and stained with 500 μL of 0.1% (*w/v*) crystal violet for 15 min at RT. The excess of crystal violet was removed by washing twice with distilled water. Absolute methanol (300 μL) was added to the wells to dissolve the crystal-violet-stained biofilms. Finally, the methanol was transferred to a 96-well microplate and the optical density (OD) values at a wavelength of 560 nm were recorded by using the SpectraMax iD3 microplate reader (Molecular Devices LLC—San Jose, CA, USA).

### 2.5. Biofilm Reduction Assay

The effects of *B. serrata* extract on *P. gingivalis* or *F. nucleatum* biofilm reduction were also measured using the crystal violet staining method [32], and adapted to our experiments from previous studies [31]. To form mature biofilms, freshly grown *P. gingivalis* or *F. nucleatum* suspensions were seeded in 24-well plates at a final concentration of $1 \times 10^7$ CFU/mL for 48 h, at 37 °C, under anaerobic conditions. Then, treatments with or without *B. serrata* extract at different final concentrations were added to the wells and incubated at 37 °C, under anaerobic conditions, for an additional 24 h. The supernatants

were then discarded, and the biomass was determined using crystal violet staining and OD values as we described above.

## 2.6. Lactate Dehydrogenase Quantification

To measure cell viability, lactate dehydrogenase (LDH) levels were measured spectrophotometrically using the CyQUANT LDH Cytotoxicity Assy Kit (cat# C20300, Thermo Fisher, Waltham, MA, USA), as we previously described [29]. Briefly, GECs ($1 \times 10^5$ cells/mL) were seeded overnight in 24-well plates (Costar, Corning—Glendale, AZ, USA). Then, the media were discarded, and the cells were treated with or without *B. serrata* extract at different final concentrations (128, 64, 32, 16, 8, 4, 2, 1, 0.5, 0.25 µg/mL) for 24 h. Lysis buffer or water was added during the last 40 min and served as internal control. After sample collection, supernatants were transferred to clear flat-bottom 96-well plates, and LDH substrate was added to all wells to be incubated for 30 min at RT, protected from light. Prior to measuring the absorbance, the reaction was stopped using the Stop Solution from the kit. Absorbance values were recorded at 490 nm and 680 nm using the SpectraMax iD3 microplate reader (Molecular Devices LLC—San Jose, CA, USA). Cells treated with lysis buffer were used as positive controls and defined as 100% cell death while cells treated with water were used to provide spontaneous cell death, and cells with no treatment were used as negative controls in the experiments.

## 2.7. Antibiotic Protection Assay

The antibiotic protection assay was performed as we previously described [28] to quantify intracellular bacterial survival after treatments with *B. serrata* extract. GECs ($3 \times 10^5$ cells/mL) were seeded in 6-well plates (Costar, Corning—Glendale, AZ, USA) overnight in media without antibiotics. Freshly grown *P. gingivalis* was added to the cells at an MOI of 100 [28,33] in OptiMEM (Gibco, Gaithersburg, MO, USA), and incubated for 2 h, at 37 °C, 5% $CO_2$. Then, the cells were washed three times with sterile prewarmed PBS and treated with metronidazole (200 µg/mL) and gentamicin (300 µg/mL) in OptiMEM for 1 h, at 37 °C, 5% $CO_2$. After incubation, the antibiotics were removed and the cells were washed three times with sterile prewarmed PBS, followed by an incubation with or without *B. serrata* extract in OptiMEM medium at different final concentrations (16, 2, 0.25 µg/mL) for an additional 21 h, at 37 °C, 5% $CO_2$. Then, the supernatants were discarded, and the cells were washed three times with sterile prewarmed PBS. Sterile distilled water was added into wells and incubated at RT for 20 min. A cell scraper was used to lyse the cells and 50 µL of each cell lysate was plated onto *Brucella* Blood Agar Plate (Anaerobe Systems, Morgan Hill, CA, USA). The plates were immediately incubated under anaerobic conditions for 10 days, at 37 °C, before colony forming units (CFU) were quantified.

## 2.8. Immunostaining for P. gingivalis

GECs ($1 \times 10^5$ cells) were seeded on 18 mm coverslips in 24-well plates to reach approximately 80% of confluence. The experimental design followed the description above for "antibiotic protection assay", in which cells were infected for 2 h, treated with antibiotics for an additional 1 h, and then incubated for additional 21 h. At the end of the experiment, infected cells on coverslips were washed three times with PBS, followed by fixation with absolute cold methanol for 10 min at RT. After three washes with PBS, the cells were permeabilized and blocked with a solution of 0.2% Triton X-100 (Sigma-Aldrich—St. Louis, MO, USA) in 5% Goat Serum (Sigma-Aldrich—St. Louis, MO, USA) and 1X PBS (Gibco—Gaithersburg, MO, USA) at 4 °C overnight. The cells were incubated with primary rabbit polyclonal antibody anti-*P. gingivalis* (cat# ANT0085, Diatheva - Cartoceto Italy) at a concentration of 1:50 prepared in 0.05% Triton X-100 in 5% Goat Serum/PBS at 4 °C overnight. After three washes with PBS, secondary goat anti-rabbit IgG (cat# A11012, Invitrogen—Carlsbad, CA, USA) at a concentration of 1:200 was prepared in 0.05% Triton X-100 in 5% Goat Serum/PBS, and was added into the wells for incubation for 2 h, protected from light, at RT. The coverslips were washed three times, counterstained, and mounted on

a slide using Vectashield Hardset Antifade Mounting Medium with DAPI (cat# H-1500, Vector Laboratories—Burlingame, CA, USA). Images were then acquired using a Nikon Eclipse 50i fluorescence microscope with an Infinity 3 camera (Nikon Instruments—Melville, NY, USA) and the Lumenera Infinity Analyze 6.3 software (Teledyne Lumenera—Ottawa, ON, Canada).

*2.9. Statistical Analysis*

Statistical analysis was performed on a Prism GraphPad (GraphPad Software, Prism 9, version 9.5.1) The results are shown in standard deviation (SD) and were analyzed using one-way Anova followed by Dunnett's test. Differences resulting in $p < 0.05$ were considered significant.

**3. Results**

*3.1. B. serrata Extract Differentially Impacts the Growth of P. gingivalis and F. nucleatum*

First, the in vitro effects of *B. serrata* extract were tested on planktonic *P. gingivalis* and *F. nucleatum* using the minimum inhibitory concentration assay. The minimum concentration of *B. serrata* extract needed to inhibit *P. gingivalis* growth was 32 µg/mL, as shown in Table 1. *F. nucleatum* growth was detected in the presence of all extract concentrations tested (512 to 0.25 µg/mL). Our data thus imply that *P. gingivalis* planktonic bacteria are more susceptible to *B. serrata* extracts than *F. nucleatum* bacteria.

**Table 1.** ***B. serrata* extract inhibits the growth of planktonic *P. gingivalis*, but not *F. nucleatum*, in vitro.** *B. serrata* extract presents antimicrobial activity against *P. gingivalis* with a MIC of 32 µg/mL, while there were no antimicrobial effects observed against *F. nucleatum* at the concentrations tested in our experiments. Results show an average of *n* = 4 experiments.

| Organisms | MIC (µg/mL) |
| --- | --- |
| *P. gingivalis* ATCC 33277 | 32 |
| *F. nucleatum* ATCC 25586 | >512 |

MIC, Minimum inhibitory concentration.

*3.2. B. serrata Extract Inhibits P. gingivalis and F. nucleatum Biofilm Formation*

Bacteria grow in the environment and on body surfaces as complex bacterial communities known as biofilms [34,35]. Biofilms provide bacteria with protection, against antibiotics and disinfectants, and a dynamic environment, which facilitates infection and enhances resistance against the immune response [34,35]. Therefore, we examined the effects of *B. serrata* extract on *P. gingivalis* or *F. nucleatum* biofilm formation. To do so, we added either *P. gingivalis* or *F. nucleatum* and different concentrations of *B. serrata* extract at the same time in 24-well plates. After 48 h, the biofilms formed at the bottom of the plate were quantified using the crystal violet staining method, as we described in the Section 2. Figure 1A shows that all concentrations tested (0.25–128 µg/mL) significantly inhibited *P. gingivalis* biofilm formation in a dose-dependent manner. Concentrations ranging from 4 to 128 µg/mL were as effective in inhibiting *P. gingivalis* biofilm formation as conventional antibiotics. However, only the higher concentration of 128 µg/mL showed a modest inhibitory effect on *F. nucleatum* biofilm formation (Figure 1B). Our positive control (penicillin/streptomycin), when added at the same time as bacteria to the wells, completely inhibited both *P. gingivalis* and *F. nucleatum* biofilm formation. These data show that *B. serrata* extract dampens *P. gingivalis* biofilm formation even at the low concentration of 0.25 µg/mL.

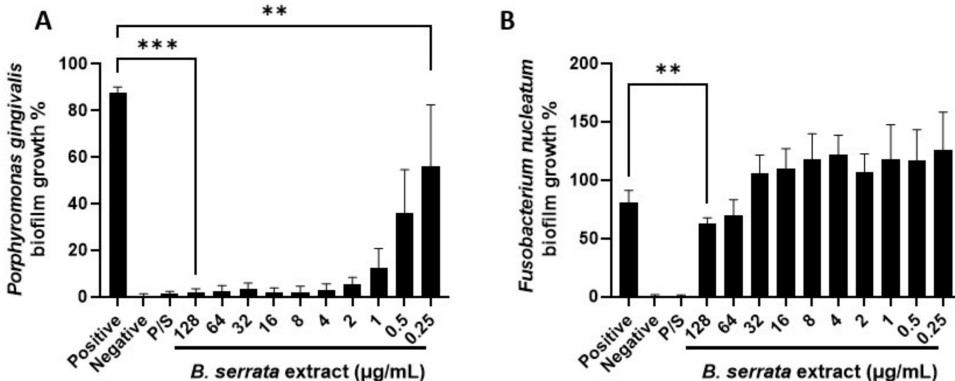

**Figure 1.** *B. serrata* **extract inhibits** *P. gingivalis* **and** *F. nucleatum* **biofilm formation in vitro.**
(**A**) *P. gingivalis* or (**B**) *F. nucleatum* were plated on 24-well plates ($1 \times 10^7$ CFU/mL) and incubated
with different concentrations of *B. serrata* extract for 48 h under anaerobic conditions. Non-adherent
bacteria were washed with phosphate-buffered saline (PBS), and adherent biofilms were fixed with
methanol and then stained with crystal violet. Dissolved biofilms were transferred to 96-well plates
and the optical density values at the wavelength of 560 nm were recorded. Graphs show biofilm
growth percentage, normalized to the positive control (only bacteria without treatments). Negative
control: only media. P/S: penicillin/streptomycin. $n = 4$. ** $p \leq 0.01$, *** $p \leq 0.001$.

### 3.3. B. serrata Extract Induces P. gingivalis Biofilm Reduction

To determine whether *B. serrata* extract could decrease established oral pathogen
biofilms, we generated *P. gingivalis* or *F. nucleatum* biofilms and then added *B. serrata*
extract at different concentrations (Figure 2). Our results show a significant dose-dependent
reduction of *P. gingivalis* biofilms (Figure 2A). The concentration of 128 µg/mL was the
most effective, and concentrations ranging from 8–128 µg/mL significantly decreased
*P. gingivalis* biofilm biomass. Interestingly, *B. serrata* extract (8–128 µg/mL) caused a more
significant decrease in *P. gingivalis* biofilm biomass compared to the conventional antibiotics,
penicillin, and streptomycin. Concentrations less than 1 µg/mL did not reduce *P. gingivalis*
biofilm biomass. Figure 2B shows that *B. serrata* extract did not affect the *F. nucleatum*
biofilm, suggesting that *F. nucleatum* biofilms are more resistant to the effects of *B. serrata*
extract compared to *P. gingivalis* biofilms.

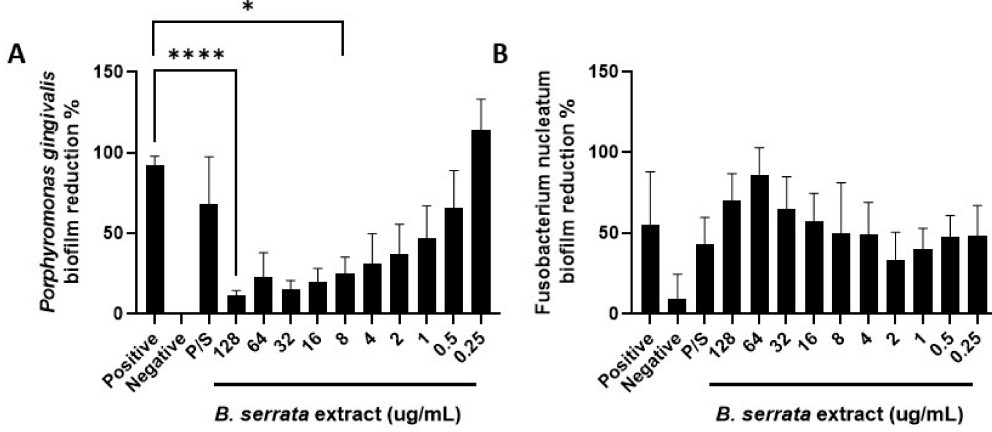

**Figure 2.** *B. serrata* **extract reduces** *P. gingivalis*, **but not** *F. nucleatum* **biofilms in vitro.**
(**A**) *P. gingivalis* or (**B**) *F. nucleatum* were plated on 24-well plates ($1 \times 10^7$ CFU/mL) and incu-
bated for 48 h under anaerobic conditions. Different concentrations of *B. serrata* extracts were added
to the biofilms and incubated for an additional 24 h. Using the crystal violet staining method, the
optical density of biofilm biomass was measured. Graphs show biofilm percentage, normalized
to the positive control (only bacteria with no treatments). Negative control: only media. P/S:
penicillin/streptomycin. $n = 3$. * $p \leq 0.05$, **** $p \leq 0.0001$.

### 3.4. B. serrata Extract at Low Doses Is Not Toxic to Human Gingival Epithelial Cells

We next examined whether the extract may have cytotoxic effects on human gingival epithelial cells. Figure 3 shows that concentrations $\geq$ 32 µg/mL are toxic to host cells, but that concentrations $\leq$ 16 µg/mL did not cause significant oral cell death. Therefore, concentrations lower than 16 µg/mL were safe to human oral cells.

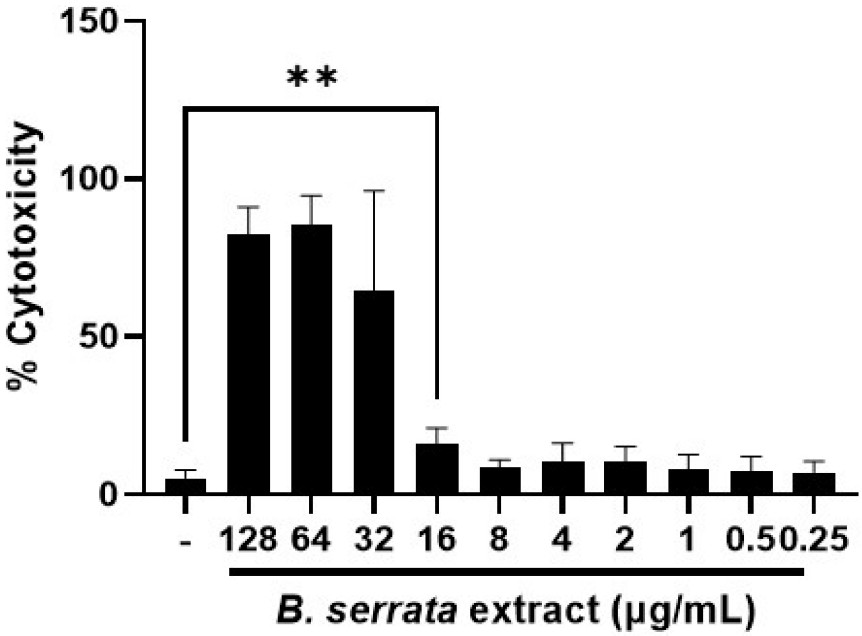

**Figure 3. Low, physiologically relevant concentrations of *B. serrata* extract are not toxic to human gingival epithelial cells.** Human gingival epithelial cells were plated on 24-well plates ($1 \times 10^5$ cells/mL) one day before treatment. Cells were then treated with different concentrations of *B. serrata* extract for 24 h. Supernatants were collected for LDH quantification. Graph shows cytotoxicity percentage, normalized to the positive control (untreated cell incubated with lysis buffer). $n = 3$. ** $p < 0.001$.

### 3.5. B. serrata Extract Decreases Intracellular P. gingivalis Infection in Human Gingival Epithelial Cells

Taken together, our data demonstrate that *B. serrata* extract is effective in decreasing *P. gingivalis*, but not *F. nucleatum*, growth and biofilm formation, and reduces biofilm biomass of *P. gingivalis*. Thus, we focused our examination of intracellular infection only on infections with *P. gingivalis*. We infected human GECs with *P. gingivalis* and then treated the cells with different concentrations of *B. serrata* extract following the antibiotic protection assay described in the Section 2. Extracellular bacteria or uninternalized bacteria attached to the host cell surface were removed during the antibiotic protection assay. This assay allowed us to detect exclusively metabolically active and viable intracellular bacteria by means of CFU quantification in blood agar plates [16,28,36–40]. Figure 4A shows that *B. serrata* extract significantly reduced the intracellular *P. gingivalis* load in a dose-dependent manner. The concentrations of 16 µg/mL and 2 µg/mL effectively killed intracellular *P. gingivalis* in infected human GECs. Our results with CFU counts in Figure 4A were confirmed by immunostaining *P. gingivalis* after infection of human GECs and treatment with *B. serrata* at 16 µg/mL (Figure 4B). Figure 4B followed the same experimental design as in Figure 4A but was analyzed by immunofluorescence microscopy to corroborate and illustrate the data in Figure 4A. Figure 4B shows representative images in which *P. gingivalis* numbers (stained in red) were significantly reduced in the group treated with *B. serrata* compared to the untreated control.

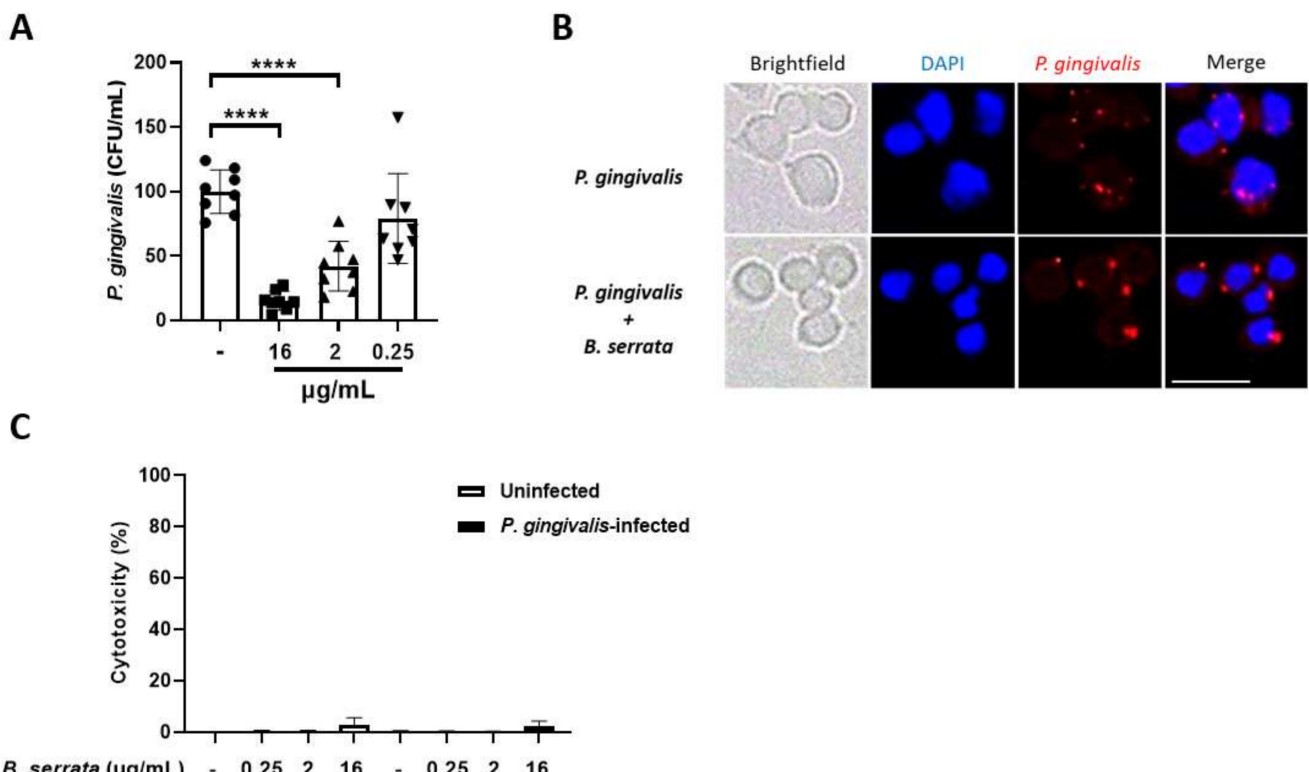

**Figure 4.** *B. serrata* **extract decreases survival of intracellular** *P. gingivalis* **in human gingival epithelial cells.** Human gingival epithelial cells were plated on 6-well plates and incubated overnight before treatment. Cells were infected, or not, with *P. gingivalis* for 2 h, followed by incubation with antibiotics (metronidazole and gentamicin) for 1 h to remove extracellular bacteria. Then, the cells were treated with or without different concentrations of *B. serrata* extract for an additional 21 h. Cells were lysed with sterile distilled water and the bacteria were plated onto blood agar plates and incubated for 10 days for CFU counts (**A**). After 21 h incubation with *B. serrata* extract, cells were used for detection of *P. gingivalis* by fluorescence microscopy (**B**), and the supernatant was used for LDH quantitation (**C**). *n* = 3. Scale bar = 100 μm. **** $p \leq 0.0001$.

To exclude the possibility that infection and *B. serrata* treatments were cytotoxic to infected GECs (which could affect the interpretation of data in Figure 4A,B), we quantified LDH from supernatants of human GECs that were infected with *P. gingivalis* and exposed them to *B. serrata* following the same experimental design as in Figure 4A,B. Figure 4C shows that neither infection nor *B. serrata* treatments induced considerable host cell death. Altogether, these results suggest that *B. serrata* selectively and significantly kills intracellular *P. gingivalis* in infected GECs, without compromising host cell integrity.

## 4. Discussion

We, among others, have reviewed data showing that *B. serrata* presents anti-cancer, anti-diabetic, and antimicrobial effects [4,10,41–43]. In this study, we explored the antimicrobial effects of *B. serrata* extract on the survival and infection of human GECs with *P. gingivalis* and *F. nucleatum*. We demonstrated that *B. serrata* extract significantly decreased *P. gingivalis* growth and biofilm formation and reduced *P. gingivalis* biofilm biomass. We also demonstrated that *B. serrata* significantly reduced *P. gingivalis* intracellular infection in human GECs, at concentrations of *B. serrata* extracts that were not toxic for GECs. Our data show, for the first time, the effects of *B. serrata* extracts on biofilms.

The overall antimicrobial effects of *B. serrata* extract on *F. nucleatum* were lower than for *P. gingivalis*. These results are consistent with previous findings from our group and others that show that *F. nucleatum* is more virulent in vitro and in vivo compared to *P. gingivalis* [18,19,28,44,45]. Additionally, both oral pathogens play different roles dur-

ing the pathogenesis of periodontitis and the formation of oral biofilms. According to Socransky et al. [12], *P. gingivalis* is a member of the red complex in the formation of sub-gingival biofilms, which is directly associated with periodontitis and is considered to be a keystone pathogen for this disease [9]. *F. nucleatum*, on the other hand, belongs to the orange complex and is believed to serve as a "bridge", essential for the connection of first colonizers to other oral pathogens of the red complex, such as *P. gingivalis* [12]. Our experiments, which corroborate previously published data [26], show that *B. serrata* did not have a direct effect on *F. nucleatum* and did not impact the growth of *F. nucleatum* planktonic bacteria or biofilm formation in vitro. Possible mechanisms involved in *F. nucleatum* resistance to *B. serrata* could be the presence of drug efflux proteins (such as the ATP-binding cassette transporter system), drug inactivation by molecules secreted by *F. nucleatum*, or target mimicry, which involves the production of molecules that bind and sequester the agonist and thus limit the effects on *F. nucleatum* bacteria [46,47]. To the best of our knowledge, no known antimicrobial resistance mechanisms in *F. nucleatum* against *B. serrata* have been described. Future studies should be focused on understanding the mechanisms involved in *F. nucleatum* resistance to the antimicrobial effects induced by *B. serrata*. Once we understand and can modulate *F. nucleatum* virulence factors, *B. serrata* would become an ideal and safe therapeutic target against both *P. gingivalis* and *F. nucleatum* during periodontitis.

Previous studies from other groups have investigated the effects of *B. serrata* and *B. sacra* on oral pathogens [26,27,48]. Raja et al. [26] screened the antibacterial activity of different boswellic acid molecules against several oral pathogens, including *P. gingivalis* and *F. nucleatum*. They showed that *P. gingivalis* was more susceptible than *F. nucleatum* to boswellic acids obtained from *B. serrata*, consistent with our own data in this study. Raja et al., focused on the effects of boswellic acids against biofilms established by *S. mutans* and *Actinomyces viscosus*, and showed that boswellic acids reduced biofilm formation by 50% [26]. Here, our study showed for the first time the effects of *B. serrata* extract on biofilm formation by *P. gingivalis* and *F. nucleatum*. Our study contributes to previous studies suggesting that *B. serrata* molecules may be used in therapies against several oral pathogens involved in different diseases, such as dental caries and periodontitis.

A more recent study evaluated the effects of *B. sacra* extracts against 12 *P. gingivalis* clinical isolates [27]. This study observed higher MIC levels (500 or 1000 µg/mL) compared to the MIC found in our study using the type-culture *P. gingivalis* 33277 strain from ATCC. Similarly to the results found in this study, the work by Attallah et al. demonstrated that frankincense extracts inhibited the biofilm formation of five *P. gingivalis* clinical isolates [27]. Even though the present study and the study by Attallah et al. [27] used frankincense extract from different sources and different tree species, the data suggest that frankincense extracts from different *Boswellia* species may have antimicrobial and antibiofilm properties against *P. gingivalis*.

Our group and others have also characterized intracellular infection by *P. gingivalis* and *F. nucleatum* in oral host cells [9,11,14,16,17,28,33,49–52]. *P. gingivalis* is an opportunistic intracellular Gram-negative pathogen able to invade several different cell types, such as gingival epithelial cells [51–53], periodontal ligament fibroblasts [54], osteoblasts and osteoclasts [55], and immune cells [28,33,56]. Due to *P. gingivalis*' ability to infect different cell types and induce inflammation, which can have systemic effects, *P. gingivalis* infection has also been associated with other non-oral health-related conditions, such as cardiovascular disease, cancer, and Alzheimer's disease [11]. The oral epithelium represents the first barrier against *P. gingivalis* and comprises the first cells to be infected before *P. gingivalis* can spread to other cells in deeper tissues, and therefore our study focused on the effects of *B. serrata* against *P. gingivalis*-infected human GECs. Previous studies had not described yet the effects of frankincense extracts against *P. gingivalis* intracellular infection in host cells. Our data showed for the first time that *B. serrata* significantly decreased intracellular infection in human GECs.

Other studies described the antimicrobial effects of frankincense extracts against airborne microbes [57], oral pathogens [26], fungi (*C. albicans* and *Malassezia furfur*), and

different types of bacteria [25]. However, the cellular and molecular mechanisms involved in the decrease of intracellular infection by *P. gingivalis* in human GECs had not yet been examined. A previous study showed that frankincense essential oil robustly modulated genome-wide gene expression in human dermal fibroblasts [58]. Given that frankincense may affect the expression of several genes in host cells, we hypothesize that *B. serrata* extracts may modulate the expression of host surface receptors involved in the recognition, uptake, and immune responses against *P. gingivalis*. Given that *P. gingivalis* traffics into endoplasmic-reticulum-rich autophagosomes for successful survival in human GECs [17], it is also possible that *B. serrata* extracts may stimulate the fusion of lysosomes to autophagosomes to contribute to *P. gingivalis* intracellular elimination in human GECs. Future studies will investigate the molecular mechanisms involved in the decrease of *P. gingivalis* intracellular infection in human GECs via *B. serrata* treatments.

Several studies describe clinical trials using frankincense extracts to treat several conditions, such as knee pain [59,60], hepatic inflammation and lipid metabolism [61], and chronic low back pain [62]. Regarding oral health, three publications report the effects of *B. serrata* extracts in the treatment of periodontitis [63–65]. Ardakani et al. showed that patients using a mouthwash consisting of five herbal extracts (including *B. serrata* extracts) improved their periodontal condition in plaque-induced gingivitis comparable to the effect of 0.2% chlorhexidine mouthwash [63]. However, because the authors used a mix of five natural products, it is not possible to conclude whether *B. serrata* extracts play a role in the improvement of clinical symptoms. Another study showed that mouthwashes containing natural products, including one with frankincense extracts, improved plaque, gingivitis, and gingival bleeding [64]. These initial studies are encouraging and suggest that clinical trials with larger sample populations and only one natural product (frankincense) are needed to confirm the effects of frankincense against periodontal disease.

We used *B. serrata* extracts to describe antimicrobial effects against oral pathogens. Boswellic acids in *B. serrata* have previously been used to explore their antimicrobial effects against oral pathogens. Among β-boswellic acid (BA), 11-keto-β-boswellic acid (KBA), acetyl-β-boswellic acid (ABA), and acetyl-11-keto-β-boswellic acid (AKBA), only BA, KBA, and AKBA showed antimicrobial effects against planktonic *P. gingivalis*, and none of the boswellic acids were effective against planktonic *F. nucleatum* [26]. In this manuscript, we described for the first time the effects of *B. serrata* extracts on biofilm formation and intracellular infection. Future studies will determine whether AKBA inhibits *P. gingivalis* biofilm formation and intracellular infection in human GECs.

Altogether, our data show that *B. serrata* extract has antimicrobial and antibiofilm effects against *P. gingivalis* in vitro. The fact that frankincense extracts have similar effects on a culture-type *P. gingivalis* strain and clinical isolates [27] reinforces the clinical relevance of studying this natural product as a potential therapeutic for periodontitis in humans. Our study, however, opens several questions that should be addressed in future studies and some limitations of the present study should be considered. The specific bioactive compound with antimicrobial effects in our extract remains to be identified. Frankincense extracts, such as *B. serrata* extracts, may have hundreds of different bioactive compounds that could be involved in antimicrobial and antibiofilm effects, such as boswellic acids [β-boswellic acid (BA), acetyl-β-boswellic acid (ABA), 3-acetyl-11-keto-β-boswellic acid (AKBA), among others] [4]. The identification of the bioactive compounds could lead to treatment that induces more specific and robust effects at lower concentrations compared to use of the whole extracts. The cellular mechanisms activated by *B. serrata* during the killing of intracellular bacteria also deserve further study.

## 5. Conclusions

In this study, we explored the antimicrobial effects of *B. serrata* extract on the survival and infection of human GECs with *P. gingivalis*. We demonstrated that *B. serrata* extract significantly decreased *P. gingivalis* growth and biofilm formation, and reduced *P. gingivalis* biofilm biomass. We also demonstrated that *B. serrata* significantly reduced *P. gingivalis* intracellular infection of human GECs without damaging the host cell. Our data show, for the first time, the effects of *B. serrata* extracts on existing biofilms.

**Author Contributions:** Conceptualization, A.C.A.M.-S., H.A., D.M.O. and C.L.C.A.-d.-S.; methodology, D.V., A.C.A.M.-S., N.Z., G.M., H.M.D. and C.L.C.A.-d.-S.; formal analysis, D.V., A.C.A.M.-S., N.Z. and C.L.C.A.-d.-S.; resources, H.A., D.M.O. and C.L.C.A.-d.-S.; writing—original draft preparation, D.V., A.C.A.M.-S., N.Z. and C.L.C.A.-d.-S.; writing—review and editing, D.M.O. and C.L.C.A.-d.-S.; supervision, H.A, D.M.O. and C.L.C.A.-d.-S.; funding acquisition, H.A., D.M.O. and C.L.C.A.-d.-S. All authors have read and agreed to the published version of the manuscript.

**Funding:** This research was funded by intramural funds from the University of the Pacific, Arthur A. Dugoni School of Dentistry to H.A., D.M.O., and C.L.C.A.-d.-S. and Start-Up Funds (D30059—Activity 101) from the University of the Pacific, Arthur A. Dugoni School of Dentistry to C.L.C.A.-d.-S.

**Institutional Review Board Statement:** Not applicable.

**Informed Consent Statement:** Not applicable.

**Data Availability Statement:** The data presented in this study are available in this article.

**Conflicts of Interest:** The authors declare no conflicts of interest.

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
