# Peer review of "Frankincense (Boswellia serrata) Extract Effects on Growth and Biofilm Formation of Porphyromonas gingivalis, and Its Intracellular Infection in Human Gingival Epithelial Cells"

_cimb, doi:10.3390/cimb46040187_

Round 1

Reviewer 1 Report

Comments and Suggestions for Authors

General comment                                 

This study investigated the effect of Frankincense extract on Porphyromonas gingivalis and Fusobacterium nucleatum biofilm and these infection into epithelial cells. That is clinically important theme to prevent and treat periodontitis.

There is a major problem with this paper that is hard to remedy. The title and the research contents and discussion are mismatched. Specifically, the investigation mainly was the antibacterial effect of Frankincense extract and the mechanisms to prevent infection of pathogenic bacteria into epithelial cells are not investigated and discussed. Nevertheless the title is "Frankincense (Boswellia serrata) extract effects on the growth, biofilm formation, and intracellular infection of Porphyromonas gingivalis and Fusobacterium nucleatum in human gingival epithelial cells"

Therefore, this manuscript is unacceptable.

Specific comment

1. Introduction section

(whole of introduction)

Written above, the novelty and hypothesis is not described.

2. Materials and methods

(line 88-90)

You write that immortalized human gingival keratinocytes (HPV-16GM) was used in line 88.

You mention the culture method for gingival epithelial cells (GECs) in line 90. Is this GECs  different to immortalized human gingival keratinocytes (HPV-16GM) ?

3. Results

(Figure.4B)

Whether or not P. gingivalis infected into epithelial cells cannot be judged in 2D image, because the image of bacterial cell (P. gingivalis) and nuclear of cells stained DAPI. How about add the 3D images?

Discussion

(line 338-339)

You write that “Future studies should be focused on understanding the mechanisms involved in F. nucleatum resistance to B. serrata. Do you have any idea and conceivable mechanisms guessed by previous study ?

(whole of discussion section)

Your discussion contains too few discussions about pathogenic bacteria and its infection to epithelial cells. Please add more discussion about intracellular infection.

Comments on the Quality of English Language

General comment                                 

This study investigated the effect of Frankincense extract on Porphyromonas gingivalis and Fusobacterium nucleatum biofilm and these infection into epithelial cells. That is clinically important theme to prevent and treat periodontitis.

There is a major problem with this paper that is hard to remedy. The title and the research contents and discussion are mismatched. Specifically, the investigation mainly was the antibacterial effect of Frankincense extract and the mechanisms to prevent infection of pathogenic bacteria into epithelial cells are not investigated and discussed. Nevertheless the title is "Frankincense (Boswellia serrata) extract effects on the growth, biofilm formation, and intracellular infection of Porphyromonas gingivalis and Fusobacterium nucleatum in human gingival epithelial cells"

Therefore, this manuscript  is unacceptable.

Specific comment

1. Introduction section

(whole of introduction)

Written above, the novelty and hypothesis is not described.

2. Materials and methods

(line 88-90)

You write that immortalized human gingival keratinocytes (HPV-16GM) was used in line 88.

You mention the culture method for gingival epithelial cells (GECs) in line 90. Is this GECs  different to immortalized human gingival keratinocytes (HPV-16GM) ?

3. Results

(Figure.4B)

Whether or not P. gingivalis infected into epithelial cells cannot be judged in 2D image, because the image of bacterial cell (P. gingivalis) and nuclear of cells stained DAPI.

How about add the 3D images?

Discussion

(line 338-339)

You write that “Future studies should be focused on understanding the mechanisms involved in F. nucleatum resistance to B. serrata. Do you have any idea and conceivable mechanisms guessed by previous study ?

(whole of discussion section)

Your discussion contains too few discussions about pathogenic bacteria and its infection to epithelial cells. Please add more discussion about intracellular infection.

Reviewer 2 Report

Comments and Suggestions for Authors

The authors did a good job of microbiology to determine the activity of Boswellia serrata extract against pathogens that cause periodontal disease. However, it seems to me that there is one major deficiency that the Authors only want to fill in the future. Chemical analysis of the extract should be included in this manuscript. Identifying at least the main components like boswellic acid seems essential. In addition, the authors could use pure boswellic acid which is available as a standard in the study. Determining its activity and content in the tested extract could largely explain the research results presented in the manuscript. 

Round 2

Reviewer 2 Report

Comments and Suggestions for Authors

The authors did not perform HPLC analysis, which does not seem difficult these days, however, they added a phytochemical element to the discussion based on literature data. I believe that the manuscript is suitable for publication, having taken into account all the suggestions of reviewers specializing in microbiology. 

Author Response

We thank the reviewer for taking the time to critically evaluate our manuscript and for the opportunity to improve our work.